# Effects of Thermal Treatment on DC Voltage-Driven Color Conversion in Organic Light-Emitting Diode

**DOI:** 10.3390/mi14010030

**Published:** 2022-12-23

**Authors:** Tae Jun Ahn, Bum Ho Choi, Jae-Woong Yu, Young Baek Kim, Yun Seop Yu

**Affiliations:** 1Department of Electrical, Electronic and Control Engineering, AISPC Laboratory, IITC, Hankyong National University, 327 Jungang-ro, Anseong-si 17579, Republic of Korea; 2Green Energy Nano Research Group, Korea Institute of Industrial Technology, 6, Cheomdangwagi-ro 208 beon-gil, Buk-gu, Gwangju 61012, Republic of Korea; 3PJPTECH, 36-2, Hagal-ro 86, Giheung-gu, Yongin-si 17096, Republic of Korea; 4Department of Advanced Materials Engineering for Information & Electronics, Kyung Hee University, Deogyeong-daro 1732, Giheung-gu, Yongin-si 17104, Republic of Korea

**Keywords:** OLED, color conversion, color tunable OLED, lighting, phthalocyanine, thermal treatment

## Abstract

A DC voltage-dependent color-tunable organic light-emitting diode (CTOLED) was proposed for lighting applications. The CTOLED consists of six consecutive organic layers: the hole injection layer, the hole transport layer (HTL), two emission layers (EMLs), a hole blocking layer (HBL), and an electron transport layer (ETL). Only one metal-free phthalocyanine (H_2_Pc) layer with a thickness of 5 nm was employed as the EML in the CTOLED on a green organic light-emitting diode (OLED) structure using tris (8-hydroxyquinoline) aluminum (III) (Alq_3_). The current density-voltage-luminance characteristics of the CTOLEDs before and after thermal treatment were characterized and analyzed. Several Gaussian peaks were also extracted by multipeak fitting analysis of the electroluminescent spectra. In the CTOLED before thermal treatment, green emission was dominant in the entire voltage range from low to high voltages, and blue and infrared were emitted simultaneously and at relatively low intensities at low and high voltages, respectively. In the CTOLED after thermal treatment, the dominant color conversion from blue to green was observed as the applied voltage increased, and the infrared emission was relatively low over the entire voltage range. By simulating the CTOLED with and without traps at the H_2_Pc interface using a technology computer-aided design simulator, we observed the following: 1. After thermal treatment, the CTOLED emitted blue light by exciton generation at the H_2_Pc–HBL interface because of the small electron transport through the H_2_Pc thin film due to the dramatic reduction of traps in the low-voltage regime. 2. In the high-voltage regime, electrons reaching the HBL were transferred to Alq_3_ by resonant tunneling in two quantum wells; thus, green light was emitted by exciton generation at the HTL–Alq_3_ interface.

## 1. Introduction

Organic light-emitting diodes (OLEDs) are rapidly replacing conventional liquid crystal displays (LCDs) in the display industry because of their unique properties, such as vivid full color, fast response times, wide viewing angles, and self-luminescence [1,2,3]. OLEDs are rapidly developing as a next-generation solid-state light source that replaces conventional incandescent and fluorescent bulbs and inorganic LED light sources [4,5,6]. The performance and lifetime of OLED lights pose potential stability issues; thus, there have been research efforts to improve the power efficiency and lifetime [7]. The characteristics of OLEDs, such as internal efficiency, outcoupling efficiency, and lifetime, have been developed to the level of commercialization [8,9]. Many studies have reported a power efficiency of 90 lm/W and a long lifetime of 20,000 h at an illuminance level of 1000 cd/m^2^ in white OLEDs (WOLEDs) [10,11,12]. Unlike inorganic LEDs, OLEDs are promising candidates for innovative lighting applications because they are thin, light, and can emit over a large area [13,14]. They can also be applied to various types of lighting, such as medical treatment, visual art, and architectural lighting. Various flexible and transparent OLEDs that provide uniform emission over a large area can be produced, which makes them suitable for various applications, such as medical treatment, visual art, and architecture lighting [15,16,17,18]. Recently, the impact of lighting environments on human emotions has received increased attention, and the need for OLED lighting that enables real-time color adjustment (color shift) has also increased [19,20]. Various types of color-tunable OLEDs (CTOLEDs) have been reported [21,22,23,24,25,26,27]. Initially, a color-tunable OLED that took advantage of the voltage-dependent change in the emission color was fabricated from a polymer blend [21,22,23]. Mechanisms that induce voltage-dependent color changes have the disadvantage of being difficult to control and resulting in undesired changes in device brightness. Subsequently, CTOLEDs that have a stacked tandem structure with multiple subpixels emitting light of different colors have been reported [24]. CTOLEDs control the emission spectrum much better but require an additional electrode for the connection between two adjacent devices to individually address each sub-pixel. This additional electrode must be made of a thin metal film layer or a transparent electrode such as indium tin oxide (ITO). Metal-thin films absorb a significant amount of light and exhibit a microcavity effect, which leads to efficiency issues. More recently, OLEDs that insert a MoO_3_ layer for charge generation and color tunability using AC voltage have also been reported [25]. An OLED for wearable devices that converts color using an exciton block layer (EBL) that manipulates the charge balance between two adjacent light-emitting layers has been reported [26]. A color-temperature-tunable white OLED (WOLED) using two undoped ultrathin (<1 nm) emitters has also been reported [27]. Existing CTOLEDs require a complicated fabrication process because they have multiple tandem structures with multiple electrodes, which can increase the manufacturing costs. In addition, CTOLEDs have efficiency and AC voltage drive issues owing to their additional electrodes. Third, CTOLEDs with a single-cell structure that has a single emission layer (EML) or multiple EMLs have also been reported [28,29,30,31]. A host-dopant-based EML is used to achieve high-efficiency WOLED, which requires fine doping of red, green, and blue, or blue and yellow fluorescent or phosphorescent dopants into the EML [28,29]. A co-host-in-double-EML device for color-tunable OLEDs based on a single tetradentate Pt(II) emitter has been reported [30]. A CTOLED with a simplified structure, including a dopant-free PN heterojunction as an EML, has been reported [31]. The reported single-cell CTOLEDs require complicated processes, such as fine doping or bandgap engineering, and additional power sources. Therefore, stable and simple DC-voltage CTOLED lighting that can be applied to various applications is required. 

In this paper, we propose a DC voltage-dependent color-tunable single-cell OLED lighting system that employs only one metal-free phthalocyanine (H_2_Pc) layer on a green OLED structure using tris (8-hydroxyquinoline) aluminum (III) (Alq_3_) as the EML. In Section 2, the structure, fabrication, and characterization of the proposed CTOLED are explained. In Section 3, the characteristics of the CTOLED before and after thermal treatment are compared, and the effects of thermal treatment are investigated. Blue and near-infrared emissions were observed simultaneously at low voltages, whereas a color change from blue to green was observed at high voltages. To physically interpret the color-tunable phenomenon of the single-cell CTOLED, a technology computer-aided design (TCAD) simulation [32] was performed. Finally, Section 4 presents the conclusions.

## 2. Experiment

Figure 1a shows a schematic of single-cell-structured CTOLED lighting devices fabricated on ITO-coated glass substrates. The CTOLED consists of six consecutive organic layers: the hole injection layer (HIL), the hole transport layer (HTL), two emission layers (EMLs), a hole blocking layer (HBL), and an electron transport layer (ETL). KHI-001, KHT-001, bathocuproine (BCP), and LG-201 were used as the HIL, HTL, HBL, and ETL, respectively. Figure 1b,c show the chemical structures of Alq_3_ and H_2_Pc, respectively, used as EMLs in the CTOLED.

An OLED whose EML layer uses only Alq_3_ can emit green light, as shown in the energy band diagram in Figure 2a [33]. In the proposed CTOLED, 5 nm-thick H_2_Pc and 30 nm-thick Alq_3_ were employed as the EMLs, and the H_2_Pc layer was prepared on the Alq_3_ layer. As shown in the energy band diagram in Figure 2b, the H_2_Pc and Alq_3_ layers can emit blue/red or green colors as electrons or holes, respectively, move through the H_2_Pc layer. It is well known that there are two energy band gaps for H_2_Pc: the Soret (B)-band and Q-band (visible). Excitons generated in the B-band and Q-band of the H_2_Pc can be created and dissipated independently of each other and can play different roles in the recombination process [34,35,36]. In Figure 2b, the red dashed line denotes the energy bandgap of the Q-band of the H_2_Pc. 

All organic materials, including H_2_Pc, were prepared using a thermal evaporator at a base pressure of 3 × 10^−6^ torr, and their deposition rate was precisely controlled by adjusting the tooling factor. An ITO with a thickness of 150 nm and sheet resistance of 10 ohm/square and Liq/aluminum with a thickness of 150 nm were used as the anode and cathode electrodes, respectively. The fabricated OLEDs were thermally treated (annealed) for 30 min at 120 °C. Typical glass encapsulation technology with getter paste has been employed to protect organic materials from water vapor and oxygen in an atmospheric environment. The emission area, which was defined during shadow mask patterning for the Al cathode formation, was 2 × 2 mm^2^. 

The electroluminescence (EL) spectra and typical current density-voltage-luminance (*J*-*V*-*L*) characteristics were tested using a *J*-*V*-*L* Measurement system M6100, which consists of a spectroradiometer CS-2000, a Keithley 2400 source meter, and a Keithley 2100 digital multimeter. 

## 3. Results and Discussions

### 3.1. Characterization of the CTOLED before Thermal Treatment

Figure 3a,b show the *J*-*V*-*L* characteristics and Commission Internationale de L’Eclairage (CIE) coordinates of the CTOLED before thermal treatment, respectively. Empty squares (□) and filled circles (●) denote current density and luminance, respectively. In Figure 3a, each voltage was marked by setting a voltage section of the current density and luminance from A to C. Points, A, B, and C denote 9, 10.5, and 12 V, respectively. In Figure 3b, the color coordinate values of the CTOLED before thermal treatment at voltages corresponding to A–C are marked in CIE coordinates. SG (0.325, 0.55) and SB (0.142, 0.14) denote the color coordinate values of single green and single blue OLEDs using Alq_3_ and H_2_Pc, respectively, as a single EML. A (0.277, 0.415) denotes the color coordinate value of a low voltage (9 V), which represents a value between green and blue. The color coordinate values from B to E are almost the same as those of SG (0.325, 0.55). Green with the same color coordinates as SG is continuously emitted at 12 V or higher, where the initial current density changes. 

Figure 4 shows the electroluminescence (EL) spectra of the CTOLED before thermal treatment at the applied voltages corresponding to A–C (Figure 3a). Figure 5 shows several Gaussian peaks extracted by multipeak fitting analysis [37] of the EL spectra at the applied voltages corresponding to A and C, respectively. In Figure 3b, the color coordinate values at A and C are (0.277, 0.415) and (0.325, 0.515), respectively. In the multipeak fitting analysis, blue, green, and infrared correspond to 450, 518, and 710 nm, respectively. In the EL spectrum at the applied voltage of 9 V (denoted as solid lines), the green and blue peaks were observed simultaneously. This shows that green and blue were emitted simultaneously from Alq_3_ and H_2_Pc (B-band), respectively. The ratio of the green, blue, and infrared peak intensities was approximately 1:10:0. That is, from the color coordinate analysis at 9 V, green was mostly emitted, while blue was simultaneously emitted at a relatively weak intensity. In the EL spectrum at the applied voltage of 12 V (denoted as dashed lines), the green and infrared spectra were observed simultaneously. This shows that green and infrared light were emitted simultaneously from Alq_3_ and H_2_Pc (Q-band), respectively. The ratio of the green, blue, and infrared peak intensities was approximately 0:19:1. Above 12 V, green was mostly emitted, while infrared was simultaneously emitted at a very weak frequency. Green emission was observed over the entire voltage range from low to high.

### 3.2. Characterization of the CTOLED after Thermal Treatment

Figure 6a,b show the *J*-*V*-*L* characteristics and CIE coordinates of the CTOLED after thermal treatment, respectively. Empty squares (□) and filled circles (●) denote current density and luminance, respectively. In Figure 6a, each voltage was marked by setting the low voltage section and peak and valley (PV) of the current density and luminance from A to E. The applied voltages of A, B, C, D, and E denote 10, 12, 15.2, 15.4, and 15.8 V, respectively. A PV current was observed in the D–E section. A PV current is well known as the current that occurs when the flow of electrons is controlled by resonant tunneling [38,39]. When the current density is greatly increased and decreased, the luminance also increases and decreases, respectively, which represents the characteristics of a typical current driving device. In Figure 6b, the color coordinate values of the CTOLED after thermal treatment at the voltages corresponding to A–E are marked in CIE coordinates. The data was divided into two groups in the blue and green regions for easy identification using the CIE coordinates: Group 1 represents points of A, B, and C, and Group 2 represents D and E. The color coordinate values from Groups 1 and 2 are almost the same as those of SB (0.142, 0.14) and SG (0.325, 0.55), respectively. As the applied voltage increases, a color shift from blue to green was observed after the first peak of the current density. The highest luminance value of blue emission was 85 cd/m^2^ at the applied voltage of C, and after color conversion, the luminance of green emission was 2.25 cd/m^2^, which was relatively lower than that before color conversion. This indicates that the color conversion from blue to green in the proposed CTOLED is driven by DC voltage.

Figure 7 shows the EL spectra of the CTOLED after thermal treatment at applied voltages from A to E (Figure 6a). At all points of the CTOLEDs after thermal treatment, a near-infrared emission spectrum was observed, whereas a weak near-infrared emission spectrum was observed at high voltages in the case of the CTOLED before thermal treatment. Figure 8 shows several Gaussian peaks extracted by multi-peak fitting analysis of the EL spectra at the applied voltages of C (15.2 V) and D (15.4 V), respectively. In the EL spectra fitted with the multi-peak analysis at the applied voltage of C (denoted as solid lines), blue emission at wavelengths of 452 and 474 nm is dominant, whereas in those at the applied voltage of D (denoted as dashed lines), green emissions at wavelengths of 518 and 568 nm are dominant.

### 3.3. Analysis of Thermal Treatment Effect

In this section, we discuss why the color conversion from blue and green emissions occurs only in the CTOLED after thermal treatment. Figure 9a,b show atomic force microscope (AFM) images of H_2_Pc thin film before and after thermal treatment, respectively. The thickness of the H_2_Pc thin film was 5 nm, the measurement area was 5 μm × 5 μm, and it was scanned in a non-contact mode. The temperature and time of thermal treatment were set to 120 °C and 30 min, respectively. Before thermal treatment, the H_2_Pc thin film with a thickness of 5 nm can be maintained in the form of a cluster or multi-grain in an unstable state, as shown in Figure 9a, and it can include many traps [40,41,42]. As shown in the schematic in Figure 10a, electrons reaching the HBL can easily be transferred to Alq_3_ through trap energy levels in the unstable H_2_Pc thin film; thus, most excitons are generated at the Alq_3_ interface, emitting a green color. Consequently, green was continuously observed over the entire voltage range. As shown in Figure 9a,b, it was confirmed that the coalescence phenomenon in which the grain boundaries of the thin film coalesce after thermal treatment was observed and the number of grains decreased; thus, the number of traps, that is, the grain boundaries of the H_2_Pc thin film, can decrease and the uniformity of the thin film was improved [40,41,42]. As shown in the schematic diagram in Figure 10b, only a small number of electrons reaching the HBL can be transferred to Alq_3_ through the stable H_2_Pc thin film, and a large number of electrons are captured in the HBL; thus, most excitons are generated at the H_2_Pc interface, emitting mainly a blue color. 

To investigate the trap effect in the H_2_Pc thin film on the emission color, TCAD simulations were carried out by dividing the state of the H_2_Pc thin film before and after thermal treatment into a case where there are traps at the H_2_Pc interface (device 1) and a case without traps (device 2), respectively. The traps were set as 1 × 10^9^ #/cm^3^ acceptor type using a Gaussian distribution with an average energy level of 3.2 eV and a capture cross section of σ = 0.6. Figure 11 shows the TCAD simulation results for the recombination positions and ratios for exciton generation at an applied voltage of 10 V with and without traps. Recombination was observed at the HTL–Alq_3_ and H_2_Pc–HBL interfaces. In the case of device 1 with traps at the H_2_Pc interface, there were recombination rates of 5 × 10^18^ #/cm^3^ and 3 × 10^14^ #/cm^3^ at the HTL–Alq_3_ and H_2_Pc–HBL interfaces, respectively, whereas in device 2 without any traps at the H_2_Pc interface, there were recombination rates of 8 × 10^5^ #/cm^3^ and 1 × 10^22^ #/cm^3^ at the HTL–Alq_3_ and H_2_Pc–HBL interfaces, respectively. This suggests that the trap can affect electron transport, as shown in the schematic in Figure 10. It can be concluded that the instability of the H_2_Pc thin film is resolved through thermal treatment; thus, blue is emitted by exciton generation at the H_2_Pc–HBL interface owing to a small electron transport through the H_2_Pc thin film by the dramatic reduction of traps in the low-voltage regime. In the high-voltage regime, electrons reaching the HBL are transferred to Alq_3_ by resonant tunneling [38,39] through the H_2_Pc thin film in two quantum wells (the HBL and Alq_3_ layers); thus, green is emitted by exciton generation at the HTL–Alq_3_ interface. The emitted color changes from blue at low voltages to green at high voltages.

Figure 12 shows the external quantum efficiency (*EQE*)–voltage (*V*) characteristics of the CTOLEDs before and after thermal treatment. The *EQE* after thermal treatment is higher than one before thermal treatment, so it can be analyzed that the stable state of the H_2_Pc thin film by thermal treatment makes a higher *EQE*. However, because the *EQE* is very low, the structure and process on the improvement of EQE must be studied in the future. 

The lifetime of thermally treated CTOLEDs was also measured and compared with that of non-treated ones. The conditions of the lifetime measurement were 38 °C of temperature and 85% relative humidity. The lifetime at the LT 95 level obtained from the non-treated one was 92 h. On the other hand, that from thermally treated CTOLED was measured to be 108 h, which was improved by 17.4%. By applying thermal treatment, it can be expected that the interface between adjacent organic emitting layers was improved by reducing traps, as mentioned before. Therefore, by employing thermal treatment on the fabricated OLED, we have observed color tunability from green to blue and white as well as an elongated lifetime.

## 4. Conclusions

We proposed and experimented on a DC voltage-dependent CTOLED that employs only one H_2_Pc layer with a thickness of 5 nm on a green OLED structure using Alq_3_ as the EML. To investigate the thermal treatment effects of the CTOLED, *J*-*V*-*L* characteristics and energy band diagrams of the CTOLEDs before and after the thermal treatment were characterized and analyzed. Several Gaussian peaks were also extracted by multi-peak fitting analysis of the EL spectra. In the CTOLED before thermal treatment, green emission was dominant in the entire voltage range from low to high voltages, and blue and infrared were emitted simultaneously and had relatively low intensities at low and high voltages, respectively. In the CTOLED after thermal treatment, the dominant color conversion from blue to green was observed as the applied voltage increased, and the infrared emission was relatively low over the entire voltage range. To investigate the reason why the CTOLED, after thermal treatment, resulted in color conversion from blue to green emissions, CTOLEDs with and without traps at the H_2_Pc interface were simulated using a TCAD simulator. Traps in the H_2_Pc layer and interface with sub-nanometer thickness can be reduced due to improved stability from thermal treatment. In the TCAD simulation, the CTOLED before thermal treatment emitted green by exciton generation at the HTL–Alq_3_ interface in the entire voltage regime because the electrons reaching the HBL could easily be transferred to Alq_3_ through trap energy levels in the unstable H_2_Pc thin film. The CTOLED after thermal treatment emitted blue by exciton generation at the H_2_Pc–HBL interface due to a small electron transport through the H_2_Pc thin film in the low voltage regime. In the high-voltage regime, electrons reaching the HBL are transferred to Alq_3_ by resonant tunneling in two quantum wells; thus, green was emitted by exciton generation at the HTL–Alq_3_ interface. In conclusion, the H_2_Pc thin film with the thickness of 5 nm can be stabilized by thermal treatment; thus, the emitted color of the proposed OLED is shifted from blue to green as the applied voltage increases and the EQE is relatively higher than before thermal treatment. However, as both EQEs before and after thermal treatment are very low, research on the improvement of EQE needs to be conducted in the future with other fabrication techniques [43]. 

## Figures and Tables

**Figure 1 micromachines-14-00030-f001:**
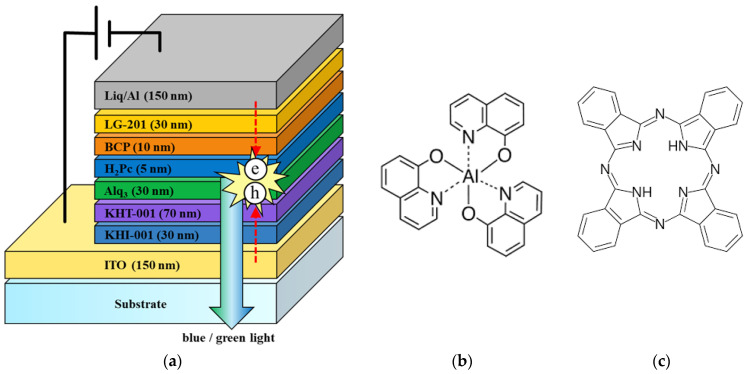
(**a**) Schematic of proposed single-cell color tunable organic light-emitting diode (CTOLED) and chemical structures of (**b**) tris (8-hydroxyquinoline) aluminum (III) (Alq_3_) and (**c**) phthalocyanine (H_2_Pc).

**Figure 2 micromachines-14-00030-f002:**
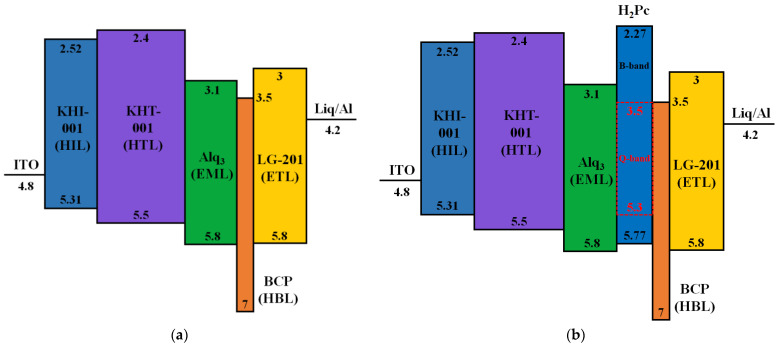
Energy band diagram of the CTOLED. (**a**) An OLED with only Alq_3_ as the EML and (**b**) a CTOLED with H_2_Pc and Alq_3_ as EMLs.

**Figure 3 micromachines-14-00030-f003:**
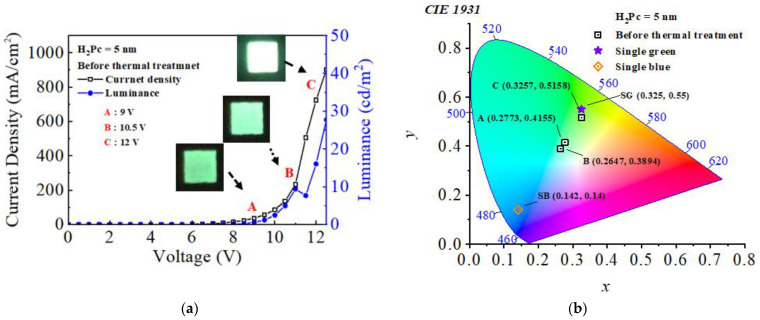
(**a**) Current density (*J*), voltage (*V*), and luminance (*L*) characteristics and (**b**) CIE coordinates of the CTOLED before thermal treatment.

**Figure 4 micromachines-14-00030-f004:**
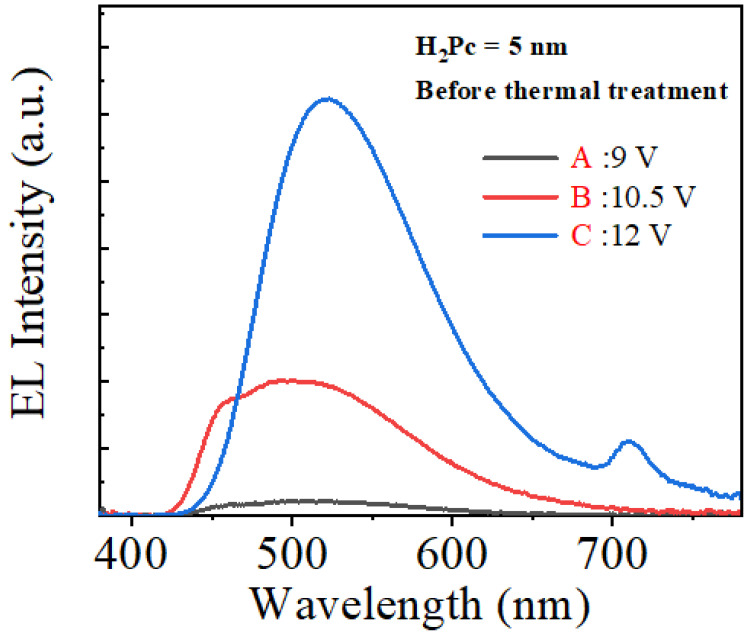
Electroluminescent (EL) spectra of the CTOLED before thermal treatment at the voltages of A, B, and C shown in Figure 3a.

**Figure 5 micromachines-14-00030-f005:**
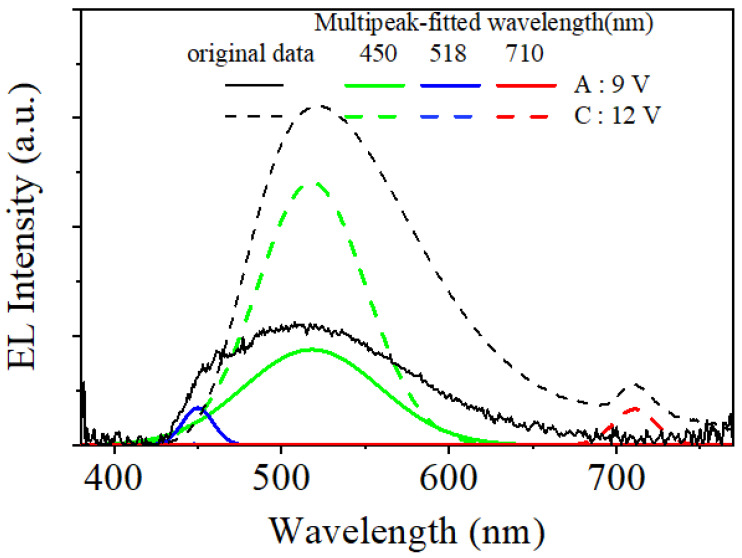
Multi-peak fit of the CTOLED before thermal treatment using a Gaussian function at applied voltages of 9 and 12 V. At the applied voltage of 9 V, original and multi-peak fitted data of ELs are shown to be amplified 10 times.

**Figure 6 micromachines-14-00030-f006:**
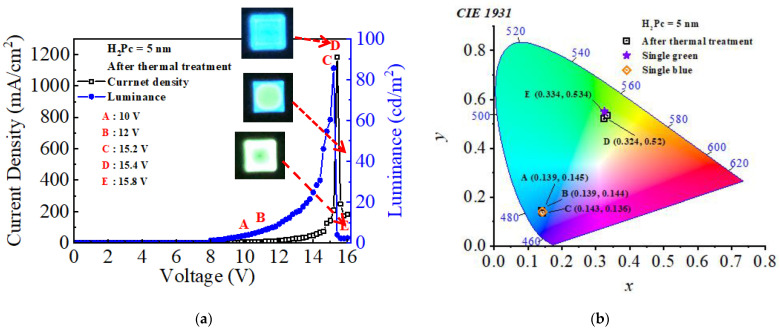
(**a**) *J*-*V*-*L* characteristics and (**b**) CIE color coordinates of the CTOLED after thermal treatment.

**Figure 7 micromachines-14-00030-f007:**
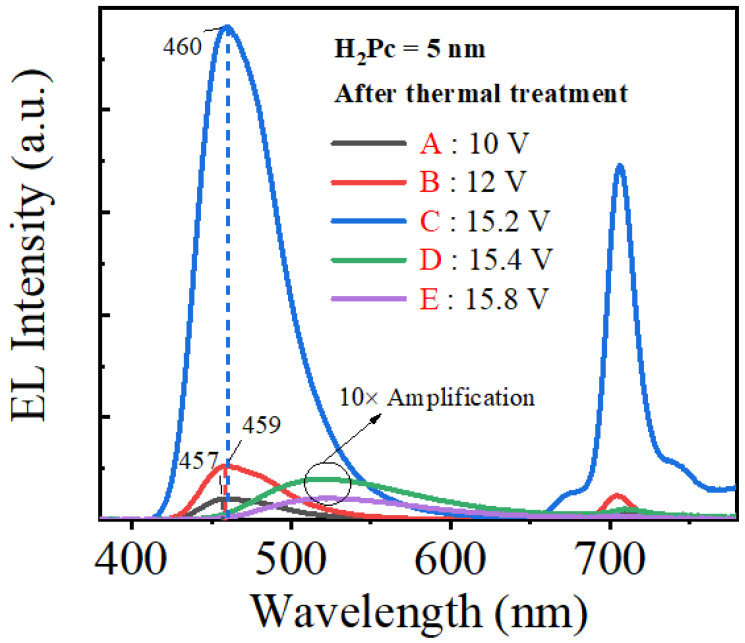
EL spectra of the CTOLED after thermal treatment at applied voltages of A, B, and C, D, and E shown in Figure 6a. At the applied voltages of D and E, ELs are shown to be amplified 10 times.

**Figure 8 micromachines-14-00030-f008:**
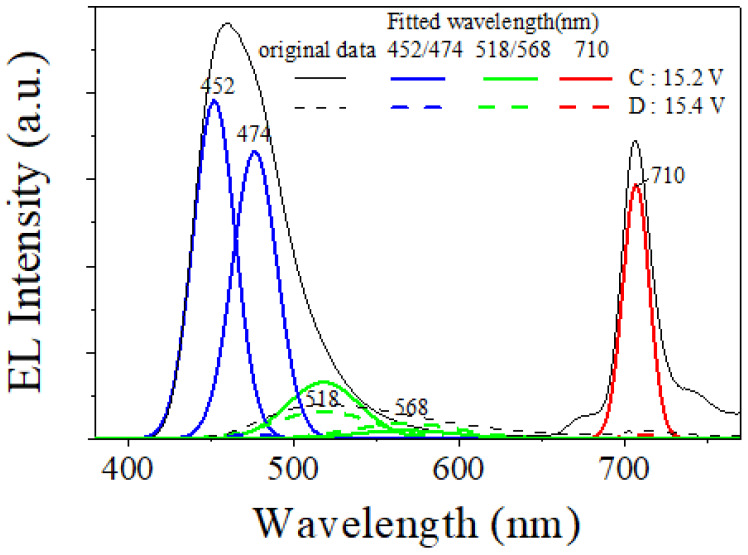
Multi-peak fit of the CTOLED after thermal treatment using a Gaussian function at an applied voltage of 15.2 and 15.4 V. At the applied voltage of 15.4 V, original and multi-peak fitted data of ELs are shown to be amplified 10 times.

**Figure 9 micromachines-14-00030-f009:**
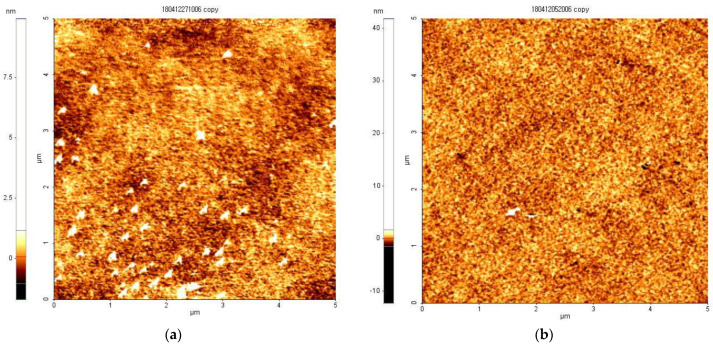
Atomic force microscope (AFM) images of H_2_Pc thin film with a thickness of 5 nm. (**a**) as-dep and (**b**) 120 °C thermal treatment for 30 min.

**Figure 10 micromachines-14-00030-f010:**
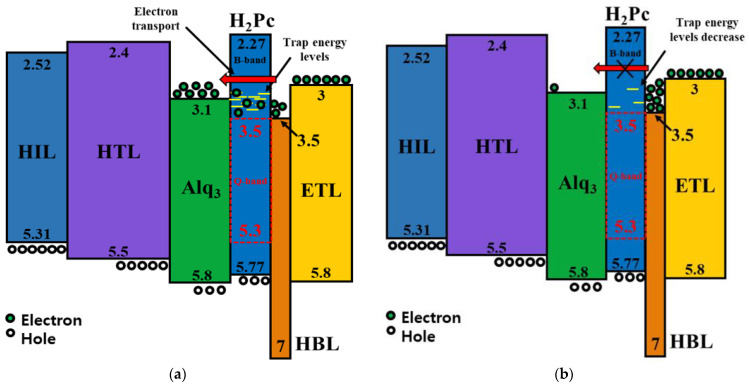
Schematic of electron transport in the case of (**a**) a large number of traps before thermal treatment and (**b**) a small number of traps after thermal treatment.

**Figure 11 micromachines-14-00030-f011:**
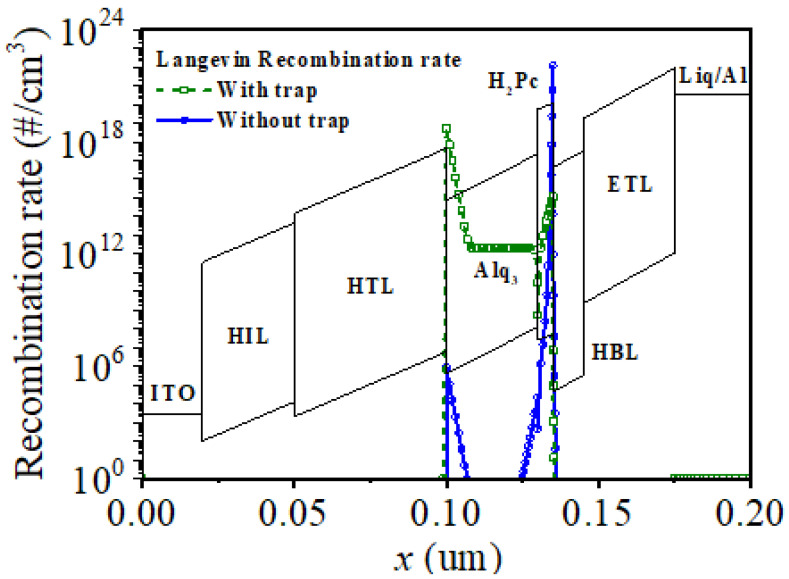
Simulation results of Langevin recombination rate at an applied voltage of 10 V with and without trap using technology computer-aided design (TCAD).

**Figure 12 micromachines-14-00030-f012:**
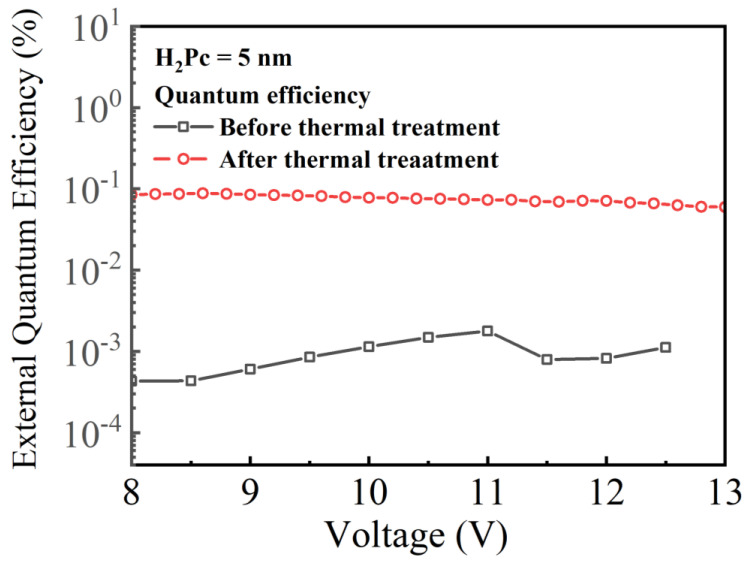
External quantum efficiency (*EQE*)-voltage (*V*) characteristics of the CTOLED before and after thermal efficiency.

## Data Availability

Not applicable.

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
