# Peer review of "Effects of Thermal Treatment on DC Voltage-Driven Color Conversion in Organic Light-Emitting Diode"

_micromachines, 2022, doi:10.3390/mi14010030_

Round 1

Reviewer 1 Report

The first, most crucial remark is concerned with the feasiblity of colour-switching effect. It has not be shown if the emission colour could be reversed by applying lower voltage. If not, then it is just degradation effect with colour change, but not the colour switching. Moreover, there is no information about the reproducibility of the effect.

The following remarks are concerned with the presented concept and the way it is called.

-The title is overloaded. Basically there are four aspects of the work: fabrication, characterization and thermal treatment effect along with the colour-switching effect itself.

-It is not clear what "single electrode" means. Obviously, a diode has two electrodes.

The article suffers from a huge number of grammar and speech mistakes that should be corrected. Here are the examples from the first page.

-"The CTOLED consists the six consecutive organic 16 layers of hole injection layer (HIL)-hole transport layer (HTL)-two emission layers (EMLs)-hole blocking layer (HBL)-electron transport layer (ETL)."

consists OF ..., no commas used in the list

-"It analyzed through technology computer-aided design (TCAD) simulation of the CTOLED with 28 and without traps at the H2Pc interface that the CTOLED after thermal treatment emits blue by 29 exciton generation at H2Pc–HBL interface due to a little electron transport through H2Pc thin film 30 by the dramatical reduction of traps at low voltage regime, and at high voltage regime, electrons 31 reaching the HBL are transferred to Alq3 by resonant tunneling in two quantum wells and thus 32 emits green by exciton generation at HTL–Alq3 interface."

it HAS BEEN analyzed or it WAS analyzed, there are actually three sentences, almost impossible to follow when merged together

-"and have been researched"

-"Organic light emitting diode (OLED) IS rapidly replacing conventional liquid crystal display (LCD) in the display industry due to THEIR unique properties such as vivid full color, fast response times, wide viewing angles, and self-luminescence [1-3]."

-"of thin and light organic materials"

first, a material cannot be thin. Thickness is not a property of material. Second, the adjective "light" in this sentence probably means "low-density", however it can be misunderstood due to light emission properties discussed in the work.

-Figure 1: The notes I=I0 and V=V0 are unnecessary and even adverse, as the voltage is varied to switch the diode and tune the color

Author Response

Please find an answer of Reviewers' comments.

Yun Seop Yu 

Reviewer 2 Report

In this manuscript, authors fabricate a DC voltage-dependent color tunable organic light emitting diode (CTOLED) with single electrode. In the CTOLED after thermal treatment, the dominant color conversion from blue to green is observed. From my personal opinion, the novelty in this paper is adequate, but some data and viewpoint seemed unconvincing enough, which are key to the novelty in this paper. So, this paper is publishable in Micromachines after necessary revisions.

Q1 The chemical structure of phthalocyanine (H2Pc) and Tris (8-hydroxyquinoline) aluminum (III) (Alq3) should be shown in one of the figures.

Q2 The role of thermal treatment is pretty interesting for OLED, I am expected to see explanations in detail.

Q3 In Figure 3a, a sharp drop in curent density can be observed at 13 V. The origin of this dorp is suggested to be clearly explained. in addition, please provide a picture of the OLED device with luminescence.

Q4 The quality of Figure 2 is poor and unclear. The authors should improve them before the publication of this work.

Q5 For light emitting diode, external quantum efficiency has always been the focus of attention (J. Mater. Chem. C, 2019, 7, 8471), so it is recommended to increase this discussion to the manuscript.

Q6 Many of the pictures are too scattered to read in the draft. It would be better if the author could combine several single graphs into a group.

Q7 The voltage interval taken in Figure 3(a) is too large, resulting in data distortion, please provide a voltage test with small intervals, such as 0.1 V or 0.2 V.

Author Response

(The authors gave the same response as above.)

Round 2

Reviewer 2 Report

In this revised edition, the author did not explain my doubts well. Although the quality of the draft has improved, there are still many awkward usages and mistakes in the manuscript. Therefore, I do not recommend publishing this manuscript in its current form.

Author Response

Please find an attached file.

Yun Seop Yu

Round 3

Reviewer 2 Report

In this revised version, the authors have carefully addressed the questions raised, but not add some key data. It is clear that the quality of this manuscript has been improved significantly. However, I have to say that the current version should be further improved.

Q1 For EL spectra shown in Figure 8, they should provide both original spectra and fitted spectra.

Q2 In the draft, the tick marks of figures without data representation on the right and top borders should be deleted. In addition, I suggest that the author should further improve the graphic quality, including font size and format.

Q3 Too many references have formatting problems, please refer to the format editing information on the Micromachines website and make corrections carefully. In addition, some important literatures on LED devices should be added (J. Mater. Chem. C, 2019, 7, 8471).

Q4 Did the operational lifetimes of the devices treated by thermal treatment method increase compared to that of the control?

Author Response

Please find one answer of Reviewer2's comment and one revised manuscript.

Sincerely,

Yun Seop Yu
